# On Valid Optimal Assignment Kernels and Applications to Graph Classification

**Nils M. Kriege**
Department of Computer Science
TU Dortmund, Germany
nils.kriege@tu-dortmund.de

**Pierre-Louis Giscard**
Department of Computer Science
University of York, UK
pierre-louis.giscard@york.ac.uk

**Richard C. Wilson**
Department of Computer Science
University of York, UK
richard.wilson@york.ac.uk

## Abstract

The success of kernel methods has initiated the design of novel positive semidefinite functions, in particular for structured data. A leading design paradigm for this is the convolution kernel, which decomposes structured objects into their parts and sums over all pairs of parts. Assignment kernels, in contrast, are obtained from an optimal bijection between parts, which can provide a more valid notion of similarity. In general however, optimal assignments yield indefinite functions, which complicates their use in kernel methods. We characterize a class of base kernels used to compare parts that guarantees positive semidefinite optimal assignment kernels. These base kernels give rise to hierarchies from which the optimal assignment kernels are computed in linear time by histogram intersection. We apply these results by developing the Weisfeiler-Lehman optimal assignment kernel for graphs. It provides high classification accuracy on widely-used benchmark data sets improving over the original Weisfeiler-Lehman kernel.

## 1  Introduction

The various existing kernel methods can conveniently be applied to any type of data, for which a kernel is available that adequately measures the similarity between any two data objects. This includes structured data like images [2, 5, 11], 3d shapes [1], chemical compounds [8] and proteins [4], which are often represented by graphs. Most kernels for structured data decompose both objects and add up the pairwise similarities between their parts following the seminal concept of convolution kernels proposed by Haussler [12]. In fact, many graph kernels can be seen as instances of convolution kernels under different decompositions [23].

A fundamentally different approach with good prospects is to *assign* the parts of one objects to the parts of the other, such that the total similarity between the assigned parts is maximum possible. Finding such a bijection is known as *assignment problem* and well-studied in combinatorial optimization [6]. This approach has been successfully applied to graph comparison, e.g., in general graph matching [9, 17] as well as in kernel-based classification [8, 18, 1]. In contrast to convolution kernels, assignments establish structural correspondences and thereby alleviate the problem of diagonal dominance at the same time. However, the similarities derived in this way are not necessarily positive semidefinite (p.s.d.) [22, 23] and hence do not give rise to valid kernels, severely limiting their use in kernel methods.

Our goal in this paper is to consider a particular class of base kernels which give rise to valid assignment kernels. In the following we use the term valid to mean a kernel which is symmetric and positive semidefinite. We formalize the considered problem: Let $[\mathcal{X}]^n$ denote the set of all $n$-element subsets of a set $\mathcal{X}$ and $\mathfrak{B}(X, Y)$ the set of all bijections between $X, Y$ in $[\mathcal{X}]^n$ for $n \in \mathbb{N}$. We study the *optimal assignment kernel* $K_{\mathfrak{B}}^k$ on $[\mathcal{X}]^n$ defined as

$$K_{\mathfrak{B}}^k(X, Y) = \max_{B \in \mathfrak{B}(X,Y)} W(B), \quad \text{where } W(B) = \sum_{(x,y) \in B} k(x, y) \tag{1}$$

and $k$ is a *base kernel* on $\mathcal{X}$. For clarity of presentation we assume $n$ to be fixed. In order to apply the kernel to sets of different cardinality, we may fill up the smaller set by new objects $z$ with $k(z, x) = 0$ for all $x \in \mathcal{X}$ without changing the result.

**Related work.** Correspondence problems have been extensively studied in object recognition, where objects are represented by sets of features often called *bag of words*. Grauman and Darrell proposed the *pyramid match kernel* that seeks to approximate correspondences between points in $\mathbb{R}^d$ by employing a space-partitioning tree structure and counting how often points fall into the same bin [11]. An adaptive partitioning with non-uniformly shaped bins was used to improve the approximation quality in high dimensions [10].

For non-vectorial data, Fröhlich et al. [8] proposed kernels for graphs derived from an optimal assignment between their vertices and applied the approach to molecular graphs. However, it was shown that the resulting similarity measure is not necessarily a valid kernel [22]. Therefore, Vishwanathan et al. [23] proposed a theoretically well-founded variation of the kernel, which essentially replaces the $\max$-function in Eq. (1) by a soft-max function. Besides introducing an additional parameter, which must be chosen carefully to avoid numerical difficulties, the approach requires the evaluation of a sum over all possible assignments instead of finding a single optimal one. This leads to an increase in running time from cubic to factorial, which is infeasible in practice. Pachauri et al. [16] considered the problem of finding optimal assignments between multiple sets. The problem is equivalent to finding a permutation of the elements of every set, such that assigning the $i$-th elements to each other yields an optimal result. Solving this problem allows the derivation of valid kernels between pairs of sets with a fixed ordering. This approach was referred to as *transitive assignment kernel* in [18] and employed for graph classification. However, this does not only lead to non-optimal assignments between individual pairs of graphs, but also suffers from high computational costs. Johansson and Dubhashi [14] derived kernels from optimal assignments by first sampling a fixed set of so-called *landmarks*. Each data point is then represented by a feature vector, where each component is the optimal assignment similarity to a landmark.

Various general approaches to cope with indefinite kernels have been proposed, in particular, for support vector machines [see 15, and references therein]. Such approaches should principally be used in applications, where similarities cannot be expressed by positive semidefinite kernels.

**Our contribution.** We study optimal assignment kernels in more detail and investigate which base kernels lead to valid optimal assignment kernels. We characterize a specific class of kernels we refer to as *strong* and show that strong kernels are equivalent to kernels obtained from a hierarchical partition of the domain of the kernel. We show that for strong base kernels the optimal assignment (i) yields a valid kernel; and (ii) can be computed in linear time given the associated hierarchy. While the computation reduces to histogram intersection similar to the pyramid match kernel [11], our approach is in no way restricted to specific objects like points in $\mathbb{R}^d$. We demonstrate the versatility of our results by deriving novel graph kernels based on optimal assignments, which are shown to improve over their convolution-based counterparts. In particular, we propose the Weisfeiler-Lehman optimal assignment kernel, which performs favourable compared to state-of-the-art graph kernels on a wide range of data sets.

## 2 Preliminaries

Before continuing with our contribution, we begin by introducing some key notation for kernels and trees which will be used later. A *(valid) kernel* on a set $\mathcal{X}$ is a function $k : \mathcal{X} \times \mathcal{X} \to \mathbb{R}$ such that there is a real Hilbert space $\mathcal{H}$ and a mapping $\phi : \mathcal{X} \to \mathcal{H}$ such that $k(x, y) = \langle \phi(x), \phi(y) \rangle$ for all $x, y$ in $\mathcal{X}$, where $\langle \cdot, \cdot \rangle$ denotes the inner product of $\mathcal{H}$. We call $\phi$ a *feature map*, and $\mathcal{H}$ a *feature space*. Equivalently, a function $k : \mathcal{X} \times \mathcal{X} \to \mathbb{R}$ is a kernel if and only if for every subset

$\{x_1, \ldots, x_n\} \subseteq \mathcal{X}$ the $n \times n$ matrix defined by $[m]_{i,j} = k(x_i, x_j)$ is p.s.d. The Dirac kernel $k_\delta$ is defined by $k_\delta(x, y) = 1$, if $x = y$ and $0$ otherwise.

We consider simple undirected graphs $G = (V, E)$, where $V(G) = V$ is the set of *vertices* and $E(G) = E$ the set of *edges*. An edge $\{u, v\}$ is for short denoted by $uv$ or $vu$, where both refer to the same edge. A graph with a unique path between any two vertices is a *tree*. A *rooted tree* is a tree $T$ with a distinguished vertex $r \in V(T)$ called *root*. The vertex following $v$ on the path to the root $r$ is called *parent* of $v$ and denoted by $p(v)$, where $p(r) = r$. The vertices on this path are called *ancestors* of $v$ and the *depth* of $v$ is the number of edges on the path. The *lowest common ancestor* $\text{LCA}(u, v)$ of two vertices $u$ and $v$ in a rooted tree is the unique vertex with maximum depth that is an ancestor of both $u$ and $v$.

## 3 Strong kernels and hierarchies

In this section we introduce a restricted class of kernels that will later turn out to lead to valid optimal assignment kernels when employed as base kernel. We provide two different characterizations of this class, one in terms of an inequality constraint on the kernel values, and the other by means of a hierarchy defined on the domain of the kernel. The latter will provide the basis for our algorithm to compute valid optimal assignment kernels efficiently.

We first consider similarity functions fulfilling the requirement that for any two objects there is no third object that is more similar to each of them than the two to each other. We will see later in Section 3.1 that every such function indeed is p.s.d. and hence a valid kernel.

**Definition 1 (Strong Kernel).** *A function* $k : \mathcal{X} \times \mathcal{X} \to \mathbb{R}_{\geq 0}$ *is called* strong kernel *if* $k(x, y) \geq \min\{k(x, z), k(z, y)\}$ *for all* $x, y, z \in \mathcal{X}$.

Note that a strong kernel requires that every object is most similar to itself, i.e., $k(x, x) \geq k(x, y)$ for all $x, y \in \mathcal{X}$.

In the following we introduce a restricted class of kernels that is derived from a hierarchy on the set $\mathcal{X}$. As we will see later in Theorem 1 this class of kernels is equivalent to strong kernels according to Definition 1. Such hierarchies can be systematically constructed on sets of arbitrary objects in order to derive strong kernels. We commence by fixing the concept of a hierarchy formally. Let $T$ be a rooted tree such that the leaves of $T$ are the elements of $\mathcal{X}$. Each inner vertex $v$ in $T$ corresponds to a subset of $\mathcal{X}$ comprising all leaves of the subtree rooted at $v$. Therefore the tree $T$ defines a family of nested subsets of $\mathcal{X}$. Let $w : V(T) \to \mathbb{R}_{\geq 0}$ be a weight function such that $w(v) \geq w(p(v))$ for all $v$ in $T$. We refer to the tuple $(T, w)$ as a *hierarchy*.

**Definition 2 (Hierarchy-induced Kernel).** *Let* $H = (T, w)$ *be a hierarchy on* $\mathcal{X}$*, then the function defined as* $k(x, y) = w(\text{LCA}(x, y))$ *for all* $x, y$ *in* $\mathcal{X}$ *is the kernel on* $\mathcal{X}$ induced *by* $H$.

We show that Definitions 1 and 2 characterize the same class of kernels.

**Lemma 1.** *Every kernel on* $\mathcal{X}$ *that is induced by a hierarchy on* $\mathcal{X}$ *is strong.*

*Proof.* Assume there is a hierarchy $(T, w)$ that induces a kernel $k$ that is not strong. Then there are $x, y, z \in \mathcal{X}$ with $k(x, y) < \min\{k(x, z), k(z, y)\}$ and three vertices $a = \text{LCA}(x, z)$, $b = \text{LCA}(z, y)$ and $c = \text{LCA}(x, y)$ with $w(c) < w(a)$ and $w(c) < w(b)$. The unique path from $x$ to the root contains $a$ and the path from $y$ to the root contains $b$, both paths contain $c$. Since weights decrease along paths, the assumption implies that $a, b, c$ are pairwise distinct and $c$ is an ancestor of $a$ and $b$. Thus, there must be a path from $z$ via $a$ to $c$ and another path from $z$ via $b$ to $c$. Hence, $T$ is not a tree, contradicting the assumption. $\qquad\square$

We show constructively that the converse holds as well.

**Lemma 2.** *For every strong kernel* $k$ *on* $\mathcal{X}$ *there is a hierarchy on* $\mathcal{X}$ *that induces* $k$.

*Proof (Sketch).* We incrementally construct a hierarchy on $\mathcal{X}$ that induces $k$ by successive insertion of elements from $\mathcal{X}$. In each step the hierarchy induces $k$ restricted to the inserted elements and eventually induces $k$ after insertion of all elements. Initially, we start with a hierarchy containing just one element $x \in \mathcal{X}$ with $w(x) = k(x, x)$. The key to all following steps is that there is a unique way to extend the hierarchy: Let $\mathcal{X}_i \subseteq \mathcal{X}$ be the first $i$ elements in the order of insertion and let $H_i = (T_i, w_i)$ be the hierarchy after the $i$-th step. A leaf representing the next element $z$ can be grafted onto $H_i$ to form a hierarchy $H_{i+1}$ that induces $k$ restricted to $\mathcal{X}_{i+1} = \mathcal{X}_i \cup \{z\}$. Let

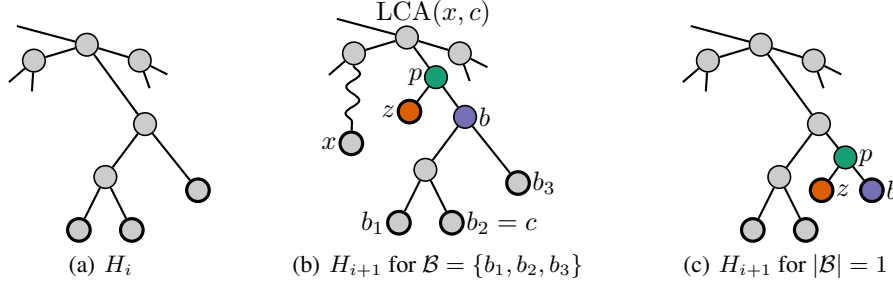

(a) $H_i$        (b) $H_{i+1}$ for $\mathcal{B} = \{b_1, b_2, b_3\}$        (c) $H_{i+1}$ for $|\mathcal{B}| = 1$

Figure 1: Illustrative example for the construction of the hierarchy on $i+1$ objects (b), (c) from the hierarchy on $i$ objects (a) following the procedure used in the proof of Lemma 2. The inserted leaf $z$ is highlighted in red, its parent $p$ with weight $w(p) = k_{\max}$ in green and $b$ in blue, respectively.

$\mathcal{B} = \{x \in \mathcal{X}_i : k(x, z) = k_{\max}\}$, where $k_{\max} = \max_{y \in \mathcal{X}_i} k(y, z)$. There is a unique vertex $b$, such that $\mathcal{B}$ are the leaves of the subtree rooted at $b$, cf. Fig. 1. We obtain $H_{i+1}$ by inserting a new vertex $p$ with child $z$ into $T_i$, such that $p$ becomes the parent of $b$, cf. Fig. 1(b), (c). We set $w_{i+1}(p) = k_{\max}$, $w_{i+1}(z) = k(z, z)$ and $w_{i+1}(x) = w_i(x)$ for all $x \in V(T_i)$. Let $k'$ be the kernel induced by $H_{i+1}$. Clearly, $k'(x, y) = k(x, y)$ for all $x, y \in \mathcal{X}_i$. According to the construction $k'(z, x) = k_{\max} = k(z, x)$ for all $x \in \mathcal{B}$. For all $x \notin \mathcal{B}$ we have $\mathrm{LCA}(z, x) = \mathrm{LCA}(c, x)$ for any $c \in \mathcal{B}$, see Fig. 1(b). For strong kernels $k(x, c) \geq \min\{k(x, z), k(z, c)\} = k(x, z)$ and $k(x, z) \geq \min\{k(x, c), k(c, z)\} = k(x, c)$, since $k(c, z) = k_{\max}$. Thus $k(z, x) = k(c, x)$ must hold and consequently $k'(z, x) = k(z, x)$. $\quad\square$

Note that a hierarchy inducing a specific strong kernel is not unique: Adjacent inner vertices with the same weight can be merged, and vertices with just one child can be removed without changing the induced kernel. Combining Lemmas 1 and 2 we obtain the following result.

**Theorem 1.** *A kernel $k$ on $\mathcal{X}$ is strong if and only if it is induced by a hierarchy on $\mathcal{X}$.*

As a consequence of the above theorem the number of values a strong kernel on $n$ objects may take is bounded by the number of vertices in a binary tree with $n$ leaves, i.e., for every strong kernel $k$ on $\mathcal{X}$ we have $|\operatorname{img}(k)| \leq 2|\mathcal{X}| - 1$. The Dirac kernel is a common example of a strong kernel, in fact, every kernel $k : \mathcal{X} \times \mathcal{X} \to \mathbb{R}_{\geq 0}$ with $|\operatorname{img}(k)| = 2$ is strong.

The definition of a strong kernel and its relation to hierarchies is reminiscent of related concepts for distances: A metric $d$ on $\mathcal{X}$ is an *ultrametric* if $d(x, y) \leq \max\{d(x, z), d(z, y)\}$ for all $x, y, z \in \mathcal{X}$. For every ultrametric $d$ on $\mathcal{X}$ there is a rooted tree $T$ with leaves $\mathcal{X}$ and edge weights, such that (i) $d$ is the path length between leaves in $T$, (ii) the path lengths from a leaf to the root are all equal. Indeed, every ultrametric can be embedded into a Hilbert space [13] and thus the associated inner product is a valid kernel. Moreover, it can be shown that this inner product always is a strong kernel. However, the concept of strong kernels is more general: there are strong kernels $k$ such that the associated kernel metric $d_k(x, y) = \|\phi(x) - \phi(y)\|$ is not an ultrametric. The distinction originates from the self-similarities, which in strong kernels, can be arbitrary provided that they fulfil $k(x, x) \geq k(x, y)$ for all $x, y$ in $\mathcal{X}$. This degree of freedom is lost when considering distances. If we require all self-similarities of a strong kernel to be equal, then the associated kernel metric always is an ultrametric. Consequently, strong kernels correspond to a superset of ultrametrics. We explicitly define a feature space for general strong kernels in the following.

### 3.1 Feature maps of strong kernels

We use the property that every strong kernel is induced by a hierarchy to derive feature vectors for strong kernels. Let $(T, w)$ be a hierarchy on $\mathcal{X}$ that induces the strong kernel $k$. We define the additive weight function $\omega : V(T) \to \mathbb{R}_{\geq 0}$ as $\omega(v) = w(v) - w(p(v))$ and $\omega(r) = w(r)$ for the root $r$. Note that the property of a hierarchy assures that the difference is non-negative. For $v \in V(T)$ let $P(v) \subseteq V(T)$ denote the vertices in $T$ on the path from $v$ to the root $r$.

We consider the mapping $\phi : \mathcal{X} \to \mathbb{R}^t$, where $t = |V(T)|$ and the components indexed by $v \in V(T)$ are

$$[\phi(x)]_v = \begin{cases} \sqrt{\omega(v)}, & \text{if } v \in P(x) \\ 0, & \text{otherwise.} \end{cases}$$

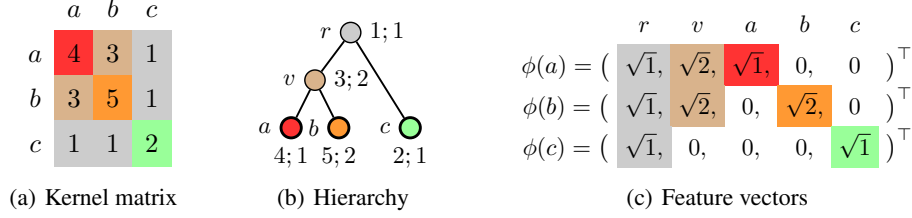

|   | a | b | c |
|---|---|---|---|
| a | 4 | 3 | 1 |
| b | 3 | 5 | 1 |
| c | 1 | 1 | 2 |

(a) Kernel matrix

(b) Hierarchy

$\phi(a) = (\ \sqrt{1},\ \sqrt{2},\ \sqrt{1},\ 0,\ 0\ )^{\top}$

$\phi(b) = (\ \sqrt{1},\ \sqrt{2},\ 0,\ \sqrt{2},\ 0\ )^{\top}$

$\phi(c) = (\ \sqrt{1},\ 0,\ 0,\ 0,\ \sqrt{1}\ )^{\top}$

(c) Feature vectors

Figure 2: The matrix of a strong kernel on three objects (a) induced by the hierarchy (b) and the derived feature vectors (c). A vertex $u$ in (b) is annotated by its weights $w(u); \omega(u)$.

**Proposition 1.** *Let $k$ be a strong kernel on $\mathcal{X}$. The function $\phi$ defined as above is a feature map of $k$, i.e., $k(x,y) = \phi(x)^{\top}\phi(y)$ for all $x, y \in \mathcal{X}$.*

*Proof.* Given arbitrary $x, y \in \mathcal{X}$ and let $c = \mathrm{LCA}(x,y)$. The dot product yields

$$\phi(x)^{\top}\phi(y) = \sum_{v \in V(T)} [\phi(x)]_v [\phi(y)]_v = \sum_{v \in P(c)} \sqrt{\omega(v)}^2 = w(c) = k(x,y),$$

since according to the definition the only non-zero products contributing to the sum over $v \in V(T)$ are those in $P(x) \cap P(y) = P(c)$. □

Figure 2 shows an example of a strong kernel, an associated hierarchy and the derived feature vectors. As a consequence of Theorem 1 and Proposition 1, strong kernels according to Definition 1 are indeed valid kernels.

## 4  Valid kernels from optimal assignments

We consider the function $K_{\mathfrak{B}}^k$ on $[\mathcal{X}]^n$ according to Eq. (1) under the assumption that the base kernel $k$ is strong. Let $(T, w)$ be a hierarchy on $\mathcal{X}$ which induces $k$. For a vertex $v \in V(T)$ and a set $X \subseteq \mathcal{X}$, we denote by $X_v$ the subset of $X$ that is contained in the subtree rooted at $v$. We define the histogram $H^k$ of a set $X \in [\mathcal{X}]^n$ w.r.t. the strong base kernel $k$ as $H^k(X) = \sum_{x \in X} \phi(x) \circ \phi(x)$, where $\phi$ is the feature map of the strong base kernel according to Section 3.1 and $\circ$ denotes the element-wise product. Equivalently, $[H^k(X)]_v = \omega(v) \cdot |X_v|$ for $v \in V(T)$. The *histogram intersection kernel* [20] is defined as $K_{\sqcap}(\mathbf{g}, \mathbf{h}) = \sum_{i=1}^{t} \min\{[\mathbf{g}]_i, [\mathbf{h}]_i\}$, $t \in \mathbb{N}$, and known to be a valid kernel on $\mathbb{R}^t$ [2, 5].

**Theorem 2.** *Let $k$ be a strong kernel on $\mathcal{X}$ and the histograms $H^k$ defined as above, then $K_{\mathfrak{B}}^k(X,Y) = K_{\sqcap}\big(H^k(X), H^k(Y)\big)$ for all $X, Y \in [\mathcal{X}]^n$.*

*Proof.* Let $(T, w)$ be a hierarchy inducing the strong base kernel $k$. We rewrite the weight of an assignment $B$ as sum of weights of vertices in $T$. Since

$$k(x,y) = w(\mathrm{LCA}(x,y)) = \sum_{v \in P(x) \cap P(y)} \omega(v), \quad \text{we have} \quad W(B) = \sum_{(x,y) \in B} k(x,y) = \sum_{v \in V(T)} c_v \cdot \omega(v),$$

where $c_v$ counts how often $v$ appears simultaneously in $P(x)$ and $P(y)$ in total for all $(x,y) \in B$. For the histogram intersection kernel we obtain

$$K_{\sqcap}(H^k(X), H^k(Y)) = \sum_{v \in V(T)} \min\{\omega(v) \cdot |X_v|, \omega(v) \cdot |Y_v|\} = \sum_{v \in V(T)} \min\{|X_v|, |Y_v|\} \cdot \omega(v).$$

Since every assignment $B \in \mathfrak{B}(X,Y)$ is a bijection, each $x \in X$ and $y \in Y$ appears only once in $B$ and $c_v \le \min\{|X_v|, |Y_v|\}$ follows.

It remains to show that the above inequality is tight for an optimal assignment. We construct such an assignment by the following greedy approach: We perform a bottom-up traversal on the hierarchy starting with the leaves. For every vertex $v$ in the hierarchy we arbitrarily pair the objects in $X_v$ and $Y_v$ that are not yet contained in the assignment. Note that no element in $X_v$ has been assigned to an element in $Y \setminus Y_v$, and no element in $Y_v$ to an element from $X \setminus X_v$. Hence, at every vertex $v$ we have $c_v = \min\{|X_v|, |Y_v|\}$ vertices from $X_v$ assigned to vertices in $Y_v$. □

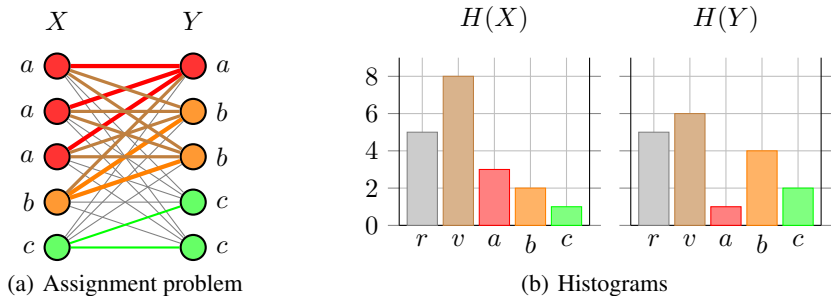

(a) Assignment problem

(b) Histograms

Figure 3: An assignment instance (a) for $X, Y \in [\mathcal{X}]^5$ and the derived histograms (b). The set $X$ contains three distinct vertices labelled $a$ and the set $Y$ two distinct vertices labelled $b$ and $c$. Taking the multiplicities into account the histograms are obtained from the hierarchy of the base kernel $k$ depicted in Fig. 2. The optimal assignment yields a value of $K_{\mathfrak{B}}^k(X, Y) = 15$, where grey, green, brown, red and orange edges have weight 1, 2, 3, 4 and 5, respectively. The histogram intersection kernel gives $K_\sqcap(H^k(X), H^k(Y)) = \min\{5, 5\} + \min\{8, 6\} + \min\{3, 1\} + \min\{2, 4\} + \min\{1, 2\} = 15$.

Figure 3 illustrates the relation between the optimal assignment kernel employing a strong base kernel and the histogram intersection kernel. Note that a vertex $v \in V(T)$ with $\omega(v) = 0$ does not contribute to the histogram intersection kernel and can be omitted. In particular, for any two objects $x_1, x_2 \in \mathcal{X}$ with $k(x_1, y) = k(x_2, y)$ for all $y \in \mathcal{X}$ we have $\omega(x_1) = \omega(x_2) = 0$. There is no need to explicitly represent such leaves in the hierarchy, yet their multiplicity must be considered to determine the number of leaves in the subtree rooted at an inner vertex, cf. Fig. 2, 3.

**Corollary 1.** *If the base kernel $k$ is strong, then the function $K_{\mathfrak{B}}^k$ is a valid kernel.*

Theorem 2 implies not only that optimal assignments give rise to valid kernels for strong base kernels, but also allows to compute them by histogram intersection. Provided that the hierarchy is known, bottom-up computation of histograms and their intersection can both be performed in linear time, while the general Hungarian method would require cubic time to solve the assignment problem [6].

**Corollary 2.** *Given a hierarchy inducing $k$, $K_{\mathfrak{B}}^k(X, Y)$ can be computed in time $\mathcal{O}(|X| + |Y|)$.*

## 5 Graph kernels from optimal assignments

The concept of optimal assignment kernels is rather general and can be applied to derive kernels on various structures. In this section we apply our results to obtain novel graph kernels, i.e., kernels of the form $K : \mathcal{G} \times \mathcal{G} \to \mathbb{R}$, where $\mathcal{G}$ denotes the set of graphs. We assume that every vertex $v$ is equipped with a categorical label given by $\tau(v)$. Labels typically arise from applications, e.g., in a graph representing a chemical compound the labels may indicate atom types.

### 5.1 Optimal assignment kernels on vertices and edges

As a baseline we propose graph kernels on vertices and edges. The *vertex optimal assignment kernel* (V-OA) is defined as $K(G, H) = K_{\mathfrak{B}}^k(V(G), V(H))$, where $k$ is the Dirac kernel on vertex labels. Analogously, the *edge optimal assignment kernel* (E-OA) is given by $K(G, H) = K_{\mathfrak{B}}^k(E(G), E(H))$, where we define $k(uv, st) = 1$ if at least one of the mappings $(u \mapsto s, v \mapsto t)$ and $(u \mapsto t, v \mapsto s)$ maps vertices with the same label only; and $0$ otherwise. Since these base kernels are Dirac kernels, they are strong and, consequently, V-OA and E-OA are valid kernels.

### 5.2 Weisfeiler-Lehman optimal assignment kernels

*Weisfeiler-Lehman kernels* are based on iterative vertex colour refinement and have been shown to provide state-of-the-art prediction performance in experimental evaluations [19]. These kernels employ the classical 1-dimensional Weisfeiler-Lehman heuristic for graph isomorphism testing and consider subtree patterns encoding the neighbourhood of each vertex up to a given distance. For a parameter $h$ and a graph $G$ with initial labels $\tau$, a sequence $(\tau_0, \ldots, \tau_h)$ of refined labels referred to as *colours* is computed, where $\tau_0 = \tau$ and $\tau_i$ is obtained from $\tau_{i-1}$ by the following procedure:

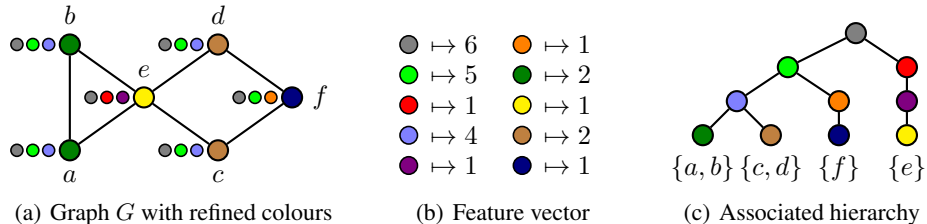

|                              |                              |                              |
| :--------------------------: | :--------------------------: | :--------------------------: |
| (a) Graph $G$ with refined colours | (b) Feature vector | (c) Associated hierarchy |

Figure 4: A graph $G$ with uniform initial colours $\tau_0$ and refined colours $\tau_i$ for $i \in \{1, \ldots, 3\}$ (a), the feature vector of $G$ for the Weisfeiler-Lehman subtree kernel (b) and the associated hierarchy (c). Note that the vertices of $G$ are the leaves of the hierarchy, although not shown explicitly in Fig. 4(c).

Sort the multiset of colours $\{\tau_{i-1}(u) : vu \in E(G)\}$ for every vertex $v$ lexicographically to obtain a unique sequence of colours and add $\tau_{i-1}(v)$ as first element. Assign a new colour $\tau_i(v)$ to every vertex $v$ by employing a one-to-one mapping from sequences to new colours. Figure 4(a) illustrates the refinement process. The *Weisfeiler-Lehman subtree kernel* (WL) counts the vertex colours two graphs have in common in the first $h$ refinement steps and can be computed by taking the dot product of feature vectors, where each component counts the occurrences of a colour, see Fig. 4(b).

We propose the *Weisfeiler-Lehman optimal assignment kernel* (WL-OA), which is defined on the vertices like OA-V, but employs the non-trivial base kernel

$$k(u, v) = \sum_{i=0}^{h} k_\delta(\tau_i(u), \tau_i(v)). \tag{2}$$

This base kernel corresponds to the number of matching colours in the refinement sequence. More intuitively, the base kernel value reflects to what extent the two vertices have a similar neighbourhood.

Let $\mathcal{V}$ be the set of all vertices of graphs in $\mathcal{G}$, we show that the refinement process defines a hierarchy on $\mathcal{V}$, which induces the base kernel of Eq. (2). Each vertex colouring $\tau_i$ naturally partitions $\mathcal{V}$ into colour classes, i.e., sets of vertices with the same colour. Since the refinement takes the colour $\tau_i(v)$ of a vertex $v$ into account when computing $\tau_{i+1}(v)$, the implication $\tau_i(u) \neq \tau_i(v) \Rightarrow \tau_{i+1}(u) \neq \tau_{i+1}(v)$ holds for all $u, v \in \mathcal{V}$. Hence, the colour classes induced by $\tau_{i+1}$ are at least as fine as those induced by $\tau_i$. Moreover, the sequence $(\tau_i)_{0 \leq i \leq h}$ gives rise to a family of nested subsets, which can naturally be represented by a hierarchy $(T, w)$, see Fig. 4(c) for an illustration. When assuming $\omega(v) = 1$ for all vertices $v \in V(T)$, the hierarchy induces the kernel of Eq. (2). We have shown that the base kernel is strong and it follows from Corollary 1 that WL-OA is a valid kernel. Moreover, it can be computed from the feature vectors of the Weisfeiler-Lehman subtree kernel in linear time by histogram intersection, cf. Theorem 2.

# 6  Experimental evaluation

We report on the experimental evaluation of the proposed graph kernels derived from optimal assignments and compare with state-of-the-art convolution kernels.

## 6.1  Method and Experimental Setup

We performed classification experiments using the $C$-SVM implementation LIBSVM [7]. We report mean prediction accuracies and standard deviations obtained by 10-fold cross-validation repeated 10 times with random fold assignment. Within each fold all necessary parameters were selected by cross-validation based on the training set. This includes the regularization parameter $C$, kernel parameters where applicable and whether to normalize the kernel matrix. All kernels were implemented in Java and experiments were conducted using Oracle Java v1.8.0 on an Intel Core i7-3770 CPU at 3.4GHz (Turbo Boost disabled) with 16GB of RAM using a single processor only.

**Kernels.**  As a baseline we implemented the *vertex kernel* (V) and *edge kernel* (E), which are the dot products on vertex and edge label histograms, respectively, where an edge label consist of the labels of its endpoints. V-OA and E-OA are the related optimal assignment kernels as described in Sec. 5.1. For the Weisfeiler-Lehman kernels WL and WL-OA, see Section 5.2, the parameter $h$ was

Table 1: Classification accuracies and standard deviations on graph data sets representing small molecules, macromolecules and social networks.

| Kernel | Data Set | | | | | | | | |
|--------|----------|--------|------|--------|----------|-----|---------|--------|--------|
| | MUTAG | PTC-MR | NCI1 | NCI109 | PROTEINS | D&D | ENZYMES | COLLAB | REDDIT |
| V | 85.4±0.7 | 57.8±0.9 | 64.6±0.1 | 63.6±0.2 | 71.9±0.4 | 78.2±0.4 | 23.4±1.1 | 56.2±0.0 | 75.3±0.1 |
| V-OA | 82.5±1.1 | 56.4±1.8 | 65.6±0.3 | 65.1±0.4 | 73.8±0.5 | 78.8±0.3 | 35.1±1.1 | 59.3±0.1 | 77.8±0.1 |
| E | 85.2±0.6 | 57.3±0.7 | 66.2±0.1 | 64.9±0.1 | 73.5±0.2 | 78.3±0.5 | 27.4±0.8 | 52.0±0.0 | 75.1±0.1 |
| E-OA | 81.0±1.1 | 56.3±1.7 | 68.9±0.3 | 68.7±0.2 | 74.5±0.6 | 79.0±0.4 | 37.4±1.8 | 68.2±0.3 | 79.8±0.2 |
| WL | **86.0**±1.7 | 61.3±1.4 | 85.8±0.2 | 85.9±0.3 | 75.6±0.4 | 79.0±0.4 | 53.7±1.4 | 79.1±0.1 | 80.8±0.4 |
| WL-OA | 84.5±1.7 | **63.6**±1.5 | **86.1**±0.2 | **86.3**±0.2 | **76.4**±0.4 | 79.2±0.4 | **59.9**±1.1 | **80.7**±0.1 | **89.3**±0.3 |
| GL | 85.2±0.9 | 54.7±2.0 | 70.5±0.2 | 69.3±0.2 | 72.7±0.6 | **79.7**±0.7 | 30.6±1.2 | 64.7±0.1 | 60.1±0.2 |
| SP | 83.0±1.4 | 58.9±2.2 | 74.5±0.3 | 73.0±0.3 | 75.8±0.5 | 79.0±0.6 | 42.6±1.6 | 58.8±0.2 | 84.6±0.2 |

chosen from $\{0, ..., 7\}$. In addition we implemented a *graphlet kernel* (GL) and the shortest-path kernel (SP) [3]. GL is based on connected subgraphs with three vertices taking labels into account similar to the approach used in [19]. For SP we used the Dirac kernel to compare path lengths and computed the kernel by explicit feature maps, cf. [19]. Note that all kernels not identified as optimal assignment kernels by the suffix OA are convolution kernels.

**Data sets.** We tested on widely-used graph classification benchmarks from different domains [cf. 4, 23, 19, 24]: MUTAG, PTC-MR, NCI1 and NCI109 are graphs derived from small molecules, PROTEINS, D&D and ENZYMES represent macromolecules, and COLLAB and REDDIT are derived from social networks.[1] All data sets have two class labels except ENZYMES and COLLAB, which are divided into six and three classes, respectively. The social network graphs are unlabelled and we considered all vertices uniformly labelled. All other graph data sets come with vertex labels. Edge labels, if present, were ignored since they are not supported by all graph kernels under comparison.

## 6.2 Results and discussion

Table 1 summarizes the classification accuracies. We observe that optimal assignment kernels on most data sets improve over the prediction accuracy obtained by their convolution-based counterpart. The only distinct exception is MUTAG. The extent of improvement on the other data sets varies, but is in particular remarkable for ENZYMES and REDDIT. This indicates that optimal assignment kernels provide a more valid notion of similarity than convolution kernels for these classification tasks. The most successful kernel is WL-OA, which almost consistently improves over WL and performs best on seven of the nine data sets. WL-OA provides the second best accuracy on D&D and ranks in the middle of the field for MUTAG. For these two data set the difference in accuracy between the kernels is small and even the baseline kernels perform notably well.

The time to compute the quadratic kernel matrix was less that one minute for all kernels and data sets with exception of SP on D&D (29 min) and REDDIT (2 h) as well as GL on COLLAB (28 min). The running time to compute the optimal assignment kernels by histogram intersection was consistently on par with the running time required for the related convolution kernels and orders of magnitude faster than their computation by the Hungarian method.

## 7 Conclusions and future work

We have characterized the class of strong kernels leading to valid optimal assignment kernels and derived novel effective kernels for graphs. The reduction to histogram intersection makes efficient computation possible and known speed-up techniques for intersection kernels can directly be applied (see, e.g., [21] and references therein). We believe that our results may form the basis for the design of new kernels, which can be computed efficiently and adequately measure similarity.

**Acknowledgments**

N. M. Kriege is supported by the German Science Foundation (DFG) within the Collaborative Research Center SFB 876 "Providing Information by Resource-Constrained Data Analysis", project A6 "Resource-efficient Graph Mining". P.-L. Giscard is grateful for the financial support provided by the Royal Commission for the Exhibition of 1851.

## Footnotes

[1]The data sets, further references and statistics are available from `http://graphkernels.cs.tu-dortmund.de`.

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
