[Reviews · NeurIPS 2016]

Reviewer 1

Summary

Optimal assignment kernels are indefinite but more naturally incorporate the assumption that similarity between two graphs is related to their structural overlap. This paper proposes a sufficient condition for positive definiteness of optimal assignment kernels by the notion of strong kernel. Then examples of valid optimal assignment kernels are derived and tested in experiments. The results indicate that the proposed approach improves classification accuracy compared to the respective related standard graph kernels.

Qualitative Assessment

Strengths: 1) The novelty of this paper is the notion of strong kernel and its equivalent relationship to kernels induced by a hierarchy, which form the basis for a valid optimal assignment kernel. This is an original and interesting contribution to knowledge, because it opens further research on a new class of valid graph kernels based on optimal assignments. In addition, the equivalence between strong kernel and kernel induced by a hierarchy is amazing, because it simplifies derivations and proofs a further results. 2) The contribution is not only significant from a theoretical but also from an empirical point of view. The experimental results are convincing. They show that valid optimal assignment kernels perform mostly better on the given datasets than the related standard graph kernels. 3) Finally, the paper is clearly presented and well structured. Limitations: No statement about how the theory applies to graphs with continuous valued node and edge attributes. In this case base kernels may not have negative values, because otherwise optimal assignments are no longer invariant under adding dummy nodes. In addition, it is unclear how useful the notion of strong kernel is for continuous valued attributes.

Confidence in this Review

2-Confident (read it all; understood it all reasonably well)


Reviewer 2

Summary

The paper puts forward a new framework for constructing kernels for structured data such as graphs that, unlike most preceding literature can represent optimal assignment of substructures (e.g. nodes or edges of graphs) whilst preserving PSD property. Optimal assignment can give sharper kernels by avoiding summing over all possible assignments many of which are suboptimal or even irrelevant. The paper shows that through using concepts of strong kernels, hierarchical representation of kernel values and histogram intersection, optimal assignment kernels can be computed. In particular, the authors show how Weisfeiler-Lehmann optimal assigment graph kernels can be computed. Experiments show that the proposed kernel has an advantage over kernels that do not consider optimal assignment.

Qualitative Assessment

The paper fills in a gap in understanding how optimal assignment can be respected in kernel computation whilst not sacrificing PSD property. The approach is very general and can thus spawn research in many directions. In graph kernel research, for me this is the most interesting paper since WL kernels were introduced. The main weakness of the technique in my view is that the kerne values will be dependent on the dataset that is being used. Thus, the effectiveness of the kernel will require a rich enough dataset to work well. In this respect, the method should be compared to the basic trick that is used to allos non-PSD similarity metrics to be used in kernel methods, namely defining the kernel as k(x,x') = (s(x,z_1),...,s(x,z_N))^T(s(x',z_1),...,s(x',z_N)), where s(x,z) is a possibly non-PSD similarity metric (e.g. optimal assignment score between x and z) and Z = {z_1,...,z_n} is a database of objects to compared to. The write-up is (understandably) dense and thus not the easiest to follow. However, the authors have done a good job in communicating the methods efficiently. Technical remarks: - it would seem to me that in section 4, "X" should be a multiset (and [\cal X]**n the set of multisets of size n) instead of a set, since in order the histogram to honestly represent a graph that has repeated vertex or edge labels, you need to include the multiplicities of the labels in the graph as well. - In the histogram intersection kernel, it think for clarity, it would be good to replace "t" with the size of T; there is no added value to me in allowing "t" to be arbitrary.

Confidence in this Review

3-Expert (read the paper in detail, know the area, quite certain of my opinion)


Reviewer 3

Summary

The paper describes an approach to construct valid (symmetric and p.s.d.) optimal assignment kernels that are built on strong base kernels. Therefore they introduce the properties of strong kernels and show, that they are induced by a hierarchy on a set of elements with increasing weights (from root to leaves) and, vice versa, every strong kernel induces such a hierarchy. Thus, on every set, by building a hierarchy, e.g. on the set of vertices of a graph, a strong kernel is induced. Using this hierarchy they derive a feature mapping and show that the scalar product of this feature mapping equals the strong kernel which therefore proofs that the strong kernel is a valid kernel. Subsequently they show that the optimal assignment can be calculated by a histogram intersection kernel where the histograms are calculated using the earlier introduced feature mapping. This allows for efficient calculation of the optimal assignment kernel once the hierarchy is built. They experimentally show that by integrating their optimal assignment kernel into existing methods one can reach higher accuracies for graph classification than state-of-the-art methods.

Qualitative Assessment

The novelty and clarity of the work seem limited. ------- Weisfeiler Lehman-OA ------- The least convincing part of the paper is the description of the Weisfeiler-Lehman OA kernel: - the second argument in equation (2) should be v not u. - the authors miss to cite Shervashidze et al. (JMLR 2011) which already presents variants of the Weisfeiler Lehman kernel, that outperform the classic one. It would have been appropriate to compare to the best of these variants. - If the proposed algorithm just counts all matching labels in the refinement sequence of all pairs of nodes, then it is completely unclear whether it actually differs from the classic Weisfeiler-Lehman kernel (which takes the linear product between the refinement sequence histograms, which is the sum over the dirac kernels on the refinement sequence of all pairs of nodes). The authors have to make this difference much clearer, define all functions/variables used (k_delta) and give an intuition will this - at best minor modification - of the WL kernel should lead to better results. ------------------ Proof Theorem 1 ------------------ The proof of Theorem 1 is not very clear. It is only shown that the histogram intersection kernel is equal to W(B) but it is not explicitly shown that this bijection B is necessarily the one maximising W(B). From my understanding this follows from the construction of the hierarchy which assures that the kernel between to nodes is maximised. However, this must be shown explicitly in the proof of Theorem 1 in order to proof that the histogram intersection kernel is equal to the optimal assignment kernel. Otherwise the proof is not complete. This missing link makes the paper also very hard to follow and requires to first understand lemma 2 in order to follow the reasoning. ------------------ Proof Lemma 2 ------------------ Fig. 1 is supposed to illustrate the proof of lemma 2, but it contains labels that are not mentioned in the text of the main article (e.g. b_1, b_2, p',c). I suggest to either modify Fig. 1 such that it fits the sketch of the proof outlined in the main article or to move the figure to the supplement. Since the construction of the hierarchy is essential for the proof of the main theorem 1, I would strongly suggest to add the complete proof to the main paper. Furthermore, there are two typos in the proof: 1. in line 128 main article, change X_{i} to \chi_{i} 2. in line 24 supplement, I suppose it should say 'since z is above LCA(B)' ------------------ Figure 3 ------------------ The caption of figure 3 is unclear to me. It would be helpful to mention, that the elements in X and Y can be represented by the hierarchy in Fig. 2. I am not sure what is meant by 'there are three vertices in X that are children of a in the hierarchy, but not represented explicitly'. As far as I understand the method, \chi = {a,b,c}, and one can build a tree T on \chi as in Fig. 2. Together with the weights w, this results in the hierarchy in Fig. 2. Now, using that X = {a,a,a,b,c} and Y = {a,b,b,c,c} and applying the definition of the histogram kernel, one can arrive at the histograms in Fig. 3b. I don't see the link between the above sentence in the figure caption. Furthermore, in Fig. 3a, I guess the edges between elements in X and Y are supposed to represent the bijections. Still, their link to the histogram kernels remains unclear to me, and it would be good to explain this detail in the caption as well. ------------------ Missing references ------------------ 1. In the paper, the Hungarian method is mentioned twice. Please add a citation or a definition. 2. The chapters 2 and 3 in the supplement are not referred to in the main article. 3. Outline difference between optimal assignment kernels and convolution kernels in more detail, or add reference

Confidence in this Review

3-Expert (read the paper in detail, know the area, quite certain of my opinion)


Reviewer 4

Summary

The paper presents a novel kernel that derives from the assignment problem. The kernel is build on simpler kernels and it provides proof that is a symmetric and s.p.d. kernel. The simpler kernel fulfills a strong inequality constraint and can be induced by hierarchy over the set. Despite its generality, the kernel has been examined under the graph comparison problem using well known datasets and outperforms in 7 out of 9 datasets the state-of-art graph kernels.

Qualitative Assessment

The paper presents a novel generalized kernel, which is build upon simpler kernels. Although the proposed method is interesting and its description quite well presented, I am a little bit concerned about an inconsistency I noticed in Definition 1. The authors define the strong kernels using a *min* condition (line 99). Two lines below (line 101), they say that strong kernel must follow a *max* condition. The *max* condition does make more sense given the explanation in line 95-97, but in the formal definition the author require the *min* condition. Which of the two conditions is the correct one? If the max condition is the correct, then proof of Lemma 1 needs to be rechecked. Furthermore, the authors evaluated their method using real world datasets and compared themselves with the state of the art methods. The experiments are well described, providing the necessary information for their evaluation. The results show that in 7 out of 9 datasets their method outperform the state-of-art graph kernel methods, demonstrating a promising method.

Confidence in this Review

2-Confident (read it all; understood it all reasonably well)


Reviewer 5

Summary

Partially ordered set(poset) like concept: "strong kernel" is introduced by regarding k(x,y) as the distance of x and y. The strong kernel is general kernel than other known definitions ( ultrametric etc ). The "poset" like structure is represented by the induced hierarchy H(T,w). The structure H(T,w) can define inner product k(x,y)=phi(x) * phi(y) using the property of the strong kernel. Given a strong kernel, the proposed method generates optimal assign kernel K_B. The assign procedure is useful especially for data with a graph structure.

Qualitative Assessment

omega and w used with similar meaning. It is better renaming omega or w. If phi cdot phi = phi^T phi then need to rewrite one of these two The experimental result shows the proposed method is quite good in accuracy and not but in computational speed. The proposed method is quit useful for data which have some graph structure. The discusstion is general. But the experiment uses delta function based kernel. The experiment does not show the usage example of the proposed method sufficiently. It is better showing example usage based gaussian kernel for example.

Confidence in this Review

2-Confident (read it all; understood it all reasonably well)


Reviewer 6

Summary

The authors of the paper proposes a classes of kernel functions to compare structural data that guarantees to be semi positive definite and that can be used to achieve optimal assignments in graph comparisons. Moreover the author proposed a method based on Weisfeiler-Lehman optimal assignment kernel comparing its performances against different graph-kernel methods and on different datasets.

Qualitative Assessment

The paper is written without major nor minor typos. The theoretical part seems to be strong and the evaluation of the method is fair. To the best of my knowledge the reference are adequate. Just a minor typo on page 3 at line 105, there is a duplicate "the". I have rated this work as follow: I rated 3 on the technical quality because the theoretical part seems to be correct and well argued and the experimental section supports the paper among datasets of different fields. This work has been rated with 3/5 because the conclusion and the discussion on the results may be extended a bit more. I rated 4 on the novelty because there are some novelty that has been not proposed in other works, i.e. the valid kernel of an optimal assignment and its fast computation in linear time. The score 4 is mainly due to the theoretical proof in the paper. I rated 3 on the potential impact because it is based on the assumption that the hierarchy and the mapping between nodes must be known a priori and this, in many real condition, is not possible to have (i.e. graph matching in which features have no label). I rated 4 on the Clarity because the paper is nicely written and the supporting figures are clear and helpful.

Confidence in this Review

1-Less confident (might not have understood significant parts)